# Effects of on-Table Extubation after Pediatric Cardiac Surgery

**DOI:** 10.3390/jcm11175186

**Published:** 2022-09-01

**Authors:** Torsten Baehner, Philipp Pruemm, Mathieu Vergnat, Boulos Asfour, Nadine Straßberger-Nerschbach, Andrea Kirfel, Michael Hamann, Andreas Mayr, Ehrenfried Schindler, Markus Velten, Maria Wittmann

**Affiliations:** 1Department of Anesthesiology and Intensive Care Medicine, University Hospital Bonn, 53127 Bonn, Germany; 2Department of Anesthesiology and Intensive Care Medicine, Stiftshospital Andernach, 56626 Andernach, Germany; 3Department of Congenital Cardiac Surgery, University Hospital Bonn, 53127 Bonn, Germany; 4Section of Pediatric Cardiac Intensive Care Medicine, University Hospital Bonn, 53127 Bonn, Germany; 5Institute for Medical Biometry, Informatics and Epidemiology, University Hospital Bonn, 53127 Bonn, Germany

**Keywords:** enhanced recovery after surgery (ERAS), on table extubation, pediatric cardiac anesthesia

## Abstract

Background: Enhanced recovery after surgery (ERAS) protocols are utilizing a multidisciplinary approach, reassessing physiology to improve clinical outcomes, reducing length of hospital stay (LOS) stay, resulting in cost reduction. Since its introduction in colorectal surgery. the concept has been utilized in various fields and benefits have been recognized also in adult cardiac surgery. However, ERAS concepts in pediatric cardiac surgery are not yet widely established. Therefore, the aim of the present study was to assess the effects of on-table extubation (OTE) after pediatric cardiac surgery compared to the standard approach of delayed extubation (DET) during intensive care treatment. Study Design and Methods: We performed a retrospective analysis of all pediatric cardiac surgery cases performed in children below the age of two years using cardiopulmonary bypass at our institution in 2021. Exclusion criteria were emergency and off pump surgeries as well as children already ventilated preoperatively. Results: OTE children were older (267.3 days vs. 126.7 days, *p* < 0.001), had a higher body weight (7.0 ± 1.6 kg vs. 4.9 ± 1.9 kg, *p* < 0.001), showed significantly reduced duration of ICU treatment (75.9 ± 56.8 h vs. 217.2 ± 211.4 h, *p* < 0.001) and LOS (11.1 ± 10.2 days vs. 20.1 ± 23.4 days; *p* = 0.001) compared to DET group. Furthermore, OTE children had significantly fewer catecholamine dependencies at 12-, 24-, 48-, and 72-h post-surgery, while DET children showed a significantly increased intrafluid shift relative to body weight (109.1 ± 82.0 mL/kg body weight vs. 63.0 ± 63.0 mL/kg body weight, *p* < 0.001). After propensity score matching considering age, weight, bypass duration, Society of Thoracic Surgeons-European Association for Cardio-Thoracic Surgery Mortality (STATS)-Score, and the outcome variables, including duration of ICU treatment, catecholamine dependencies, and hospital LOS, findings significantly favored the OTE group. Conclusion: Our results suggest that on-table extubation after pediatric cardiac surgery is feasible and in our cohort was associated with a favorable postoperative course.

## 1. Introduction

Individualized, high-quality, and resource aware peri-operative care is outcome relevant in the modern area of pediatric cardiac surgery [1,2]. Enhanced recovery after surgery (ERAS) protocols have been introduced beginning in colorectal surgery more that 2 decades ago and have shown to improve postoperative outcome while reducing length of hospital stay (LOS) [3]. Since then, this concept has widely been accepted to improve postoperative outcomes especially regarding the prevention of postoperative cognitive dysfunction and reduction of cardiac and pulmonary complications, as well as nausea and vomiting [4]. At the same time, the first ERAS protocols were published in adult cardiac surgery [5]. In addition to these obvious advantages, when children are undergoing surgery, reduced exposure to anesthetics, especially in the neonatal period, has additional significant benefits. The neurotoxic effects of anesthetics in the developing brain have been acknowledged for a long time [6,7,8]. In this context, any extended anesthesia duration and accumulation of anesthetics should be prevented [9]. In particular, within the first two years of extrauterine brain development, when most surgical procedures for congenital cardiac defects are performed, the administration of various anesthetics has been shown to impair neurological development and accelerate cerebral apoptosis [6,10,11]. Although, a single exposure to anesthetics within the first 36 months of life has not been shown to cause long-term cognitive impairment in otherwise healthy children, experimental animal studies suggest that protracted exposures may adversely affect cerebral development [12,13,14]. In addition to the neurotoxicity aspect noted previously, rapid termination of anesthesia after surgery, and thus reduction of controlled ventilation duration, in children with passive pulmonary perfusion has substantial hemodynamic benefits [15]. The shift from controlled to a spontaneous or even assisted ventilation mode in children with univentricular cardiac physiology leads to a major improvement of pulmonary blood flow and oxygenation [15,16]. During controlled ventilation, the pulmonary preload needs to be kept at a high level to ensure passive pulmonary blood flow, accepting significant amounts of volume intake with all the associated negative consequences of fluid overload. These facts have been previously acknowledged. However, on-table extubation (OTE) after pediatric cardiac surgery is not the standard procedure and regularly performed at some centers only [17]. Reasons for commonly performed delayed extubation (DET) after transfer to intensive care unit are various including caution of early postoperative bleeding or concern about difficult to assess and manage combined respiratory and circulatory insufficiency [18,19].

Recently, there has been a discussion regarding whether early extubation may also be beneficial in pediatric patients, potentially improving clinical outcomes, including a 30%–50% reduction in LOS resulting not only in decreased health care expenses, but most notably a reduction of postoperative morbidity [4]. However, standards differ between centers and OTE is performed at a few institutions only [17]. At our center for pediatric cardiac surgery, early OTE has been established. However, both OTE and DET after admission to ICU are still performed. In each case the team makes a joint decision of OTE or DET on a patient-individualized basis including the course of the surgical procedure and hemodynamic situation. The aim of the present retrospective single center analysis was to assess the efficacy and safety of OTE in pediatric cardiac surgical patients. Therefore, we retrospectively analyzed the records of 152 pediatric cardiac surgery cases.

## 2. Materials and Methods

In accordance to the Declaration of Helsinki and §15 of the Medical Association Nordrheins’ professional code of conduct, we performed a retrospective analysis of all pediatric cardiac surgery cases on patients less than or equal to 24 months of age and using cardiopulmonary bypass at the University Medical Center Bonn, Germany, between 1 January 2021 and 31 December 2021. Data analysis was based on medical records and electronic patient file including electronic patient data management system. Inclusion criteria were: age less than or equal to 24 months, elective surgery, and use of cardiopulmonary bypass. Exclusion criteria were: re-surgery, insufficient or missing of relevant data, emergency surgery, and prior to surgery already ventilated children. Hence, 152 patients were included in the study. A total of 7 patients had to be excluded. In 5 patients, the exact time of extubation could not be determined. In one case, the child was already admitted to our center on mechanical ventilation. In one case, despite initial planning of a cardiopulmonary bypass, the surgery was performed without the usage of cardiopulmonary bypass. Finally, 145 cases were included for further analysis. Total ventilation time for DET was defined as intraoperative ventilation and postoperative mechanical ventilation duration in the intensive care unit until extubation. Early extubation was defined as OTE in the operating theatre after surgery. Cumulative fluid balance and catecholamine dosage related to body weight at the end of surgery and on postoperative days 1–3, as well as duration of intensive care treatment and length of hospital stay (LOS) were evaluated. The vasoactive-inotropic score (VIS) was used to assess and evaluate cumulative catecholamine therapy [20]. The duration of catecholamine therapy was defined as the time from the start of surgery to the end of any inotropes or vasopressors in the OR or ICU. Postoperative fluid balance was related to preoperative body weight in kilograms and expressed in ml/kg. There are three models for stratification of complexity used with similar discriminatory capacity [21]. The present study used the Society of Thoracic Surgeons-European Association for Cardio-Thoracic Surgery (STAT)-score to assess case complexity and risk for mortality associated with congenital heart surgery.

### 2.1. Anesthesia and Sedation Protocol

General anesthesia was induced by inhalational of sevoflurane or intravenous injection of propofol and rocuronium (0.3 mg/kg for intubation). After endotracheal intubation, using an age adjusted micro-cuff tube, balanced anesthesia was maintained according to a standardized protocol using sevoflurane (minimum alveolar concentration 0.5%) and remifentanil (10–20 mcg/kg/h). Standard monitoring was used, including electrocardiogram (ECG), non-invasive blood pressure, O2 saturation, temperature, invasive blood pressure, near-infrared spectroscopy (NIRS), and bi-spectral index (BIS). In the OTE group, patients received piritramide (0.2 mg/kg) before propofol and remifentanil infusions were terminated at the end of the surgery. Subsequent analgesia was performed using piritramide (0.05 mg/kg every 4 h) and paracetamol (10 mg/kg every 6 h). Medications were given oral beginning on postoperative day 2. For the DET group, at the end of the surgery, patients were sedated during transfer and admission on ICU using remifentanil (10–20 mcg/kg/h) and propofol (5–10 mg/kg/h). Piritramide (0.05 mg/kg every 4 h) and paracetamol (10 mg/kg every 6 h) were started before extubation and termination of remifentanil and propofol infusions. Sufentanil (1–3 mcg/kg/h), midazolam 0.1–0.3 mg/kg/h), and ketamin (1 mg/kg) were given patients requiring prolonged sedation. Additional analgesics were given if patients showed signs of pain or stress.

### 2.2. Statistical Analyses

Statistical analyses were performed using SPSS 28 (SPSS Inc., Chicago, IL, USA) and the statistical programming environment R version 4.1.2 (Foundation for Statistical Computing, Vienna, Austria). Descriptive statistics are presented as mean ± standard deviation (SD) for symmetric continuous variables or median with inter-quartile range (IQR) for skewed variables. In the exploratory data analysis, differences between treatment groups (OTE vs. DET) were determined using two-sample *t*-test or non-parametric rank-based Mann-Whitney tests for skewed data. To account for potential confounders affecting the treatment decision and potential outcomes, propensity score-matching was performed based on logistic regression with the variables age, weight, complexity of performed procedure using STAT-Sore as well as duration of cardiopulmonary bypass. Based on nearest neighbor matching, for every OTE patient one control (DET) patient was selected via the corresponding propensity score leading to an equal number of controls and cases [22]. Performance of the matching procedure was assessed graphically via group differences in the matching variables before and after the approach was performed. After matching, differences between treatment groups were assessed using Wilcoxon signed rank test for paired differences. As our approach represents an exploratory analysis of observational data, we refrained from adjusting the typical two-sided significance level of 0.05 for multiple testing [23].

## 3. Results

A total of 43 of the 145 patients (29.6%) were OTE in the OR while 102 (70.3%) were DET during intensive care treatment. Ventilation duration was significant shorter in patients that were OTE compared to DET patients (5.7 *±* 1.5 OTE vs. 103.3 *±* 97.1 DET hours mean *±* SD, *p* < 0.0001). Furthermore, reintubation during the course of postoperative intensive care treatment occurred in one OTE patient due to re-surgery for bleeding complication and 10 DET patients all due to respiratory complications.

Demographics and intraoperative characteristics differed significantly between OTE and DET group. On table extubated children were significantly older (267 *±* 163 OTE vs. 127 *±* 148 DET days mean *±* SD, *p* < 0.0001), had a higher body weight (7.0 *±* 1.6 OTE vs. 4.9 *±* 1.9 DET kg mean *±* SD, *p* < 0.0001), and were larger (68 *±* 8 OTE vs. 59 *±* 8 DET cm mean *±* SD, *p* < 0.0001) compared to children that were DET during intensive care treatment (Figure 1a–c).

Furthermore, the duration of surgery was significantly shorter in the OTE group compared to DET group (237 *±* 61 OTE vs. 323 *±* 81 DET min mean *±* SD, *p* < 0.0001). Similarly, cardiopulmonary bypass duration was significantly shorter in the OTE group compared to DET group (119 *±* 54 OTE vs. 162 *±* 62 DET min mean *±* SD, *p* < 0.0001).

Patients in the OTE group received significantly fewer RBC transfusions (35 *±* 16 OTE vs. 75 *±* 39 DET ml/kg, mean *±* SD, *p* < 0.0001) and fewer crystalloid fluids substitutions (63 *±* 563 OTE vs. 109 *±* 82 DET ml/kg, mean *±* SD, *p* < 0.0001) compared to DET group relative to absolute body weight. After the intraoperative fluid loading, patients in the DET group showed greater negative fluid balances during the postoperative course (Figure 2).

Catecholamine dosages were not different between groups prior to surgery (1.5 *±* 2.7 OTE vs. 3.1 *±* 4.8 DET VIS score, mean *±* SD, *p* = 0.106). However, catecholamine dosages represented by the VIS score were significantly lower in OTE compared to DET patients during the postoperative period (Figure 3). Furthermore, weaning from catecholamine therapy was much faster in OTE patients compared to DET patients (48.6 *±* 52.9 OTE vs. 155.6 *±* 147.7 DET h, mean *±* SD, *p* < 0.0001). Length of hospital stay (LOS) (11 *±* 10 OTE vs. 20 *±* 23 DET d, mean *±* SD, *p* = 0.001) and duration of intensive care treatment (76 *±* 57 OTE vs. 217 *±* 211 DET h, mean *±* SD, *p* < 0.0001) were significantly shorter in OTE patients compared to DET patients (Figure 4).

Perioperative inflammatory markers were assessed beginning prior to surgery and daily until day three. Leucocytes and CRP (C-reactive protein) were not different between groups prior to surgery. Leucocytes and CRP increased in both groups after surgery. However, there was no difference observed between DET or OTE during the consecutive postoperative course until day three (Table 1).

Pathologies differed significantly between OTE and DET groups with some procedures only extubated after admission to ICU (Supplemental 1). The impact of the type of surgical procedure and/or the degree of complexity of the surgical intervention on the decision of OTE or DET was evaluated on the basis of the STATS-score. Children exhibiting a higher STATS-score were significantly more likely to be DET during intensive care treatment (Figure 5).

To evaluate the impact of potential confounders on the postoperative outcome we performed propensity score matching followed by “nearest neighbor” procedures for similar cases. We matched general demographics such as age and weight, and indicators of surgery such as complexity of performed procedure using STAT-Score as well as duration of cardiopulmonary bypass. Duration of intensive care treatment and catecholamine therapy remained significantly shorter in OTE patients compared to DET patients after propensity score matching (Figure 6), while the difference regarding general LOS no longer reached statistical significance (*p* = 0.095).

## 4. Discussion

On-table extubation is nowadays an established part of enhanced recovery protocols after pediatric cardiac surgery [17], but the frequency of performing OTE significantly differs between pediatric cardiac surgery centers even with comparable patient collectives and cases [24]. On-table extubation has been performed with an increasing frequency at our institution in recent years. However, which patients benefit most from early extubation, and in which patients an early extubation strategy leads to increased risk for perioperative complications has not been objectively evaluated [18,19]. Therefore, the aim of this study was to assess, whether on table extubation is a safe procedure and to evaluate potential benefit from early extubation.

To evaluate the safety and potential benefits of OTE, we retrospectively reviewed infants up to 24 months of age that received pediatric cardiac surgery using cardiopulmonary bypass. Using these inclusion criteria, we have already selected a higher-risk collective, especially due to early age and usage of extracorporeal circulation. Our data indicate that successful early extubation was predominantly performed in infants with an advanced age, body weight, and height as well as lower cardiopulmonary bypass durations. This observation is conclusive because the youngest and smallest children that receive the most complex procedures with extended surgical duration are exposed to longest cardio pulmonary bypass duration and represent the highest risk population. These results are is in line with the work of Winch et al., who analyzed 416 children that underwent congenital cardiac surgery within the first year of life to assess predisposing factors for a successful early extubation [25]. Early extubation in the operating room could successfully be performed in 56% of the children, whereas 10% required reintubation, most frequently within the first 24 h. Although the infants were younger than the patients in our study, Winch et al. reported a higher extubation rate compared to our data. However, this was associated with an increased re-intubation rate that was also in contrast to our results. Our reported OTE rate is comparable to Varghese et al., who reported an OTE rate of 30.4% in a group of 148 patients after cardiac surgery with a mean age of seven days [26]. Also consistent with our data, this study demonstrated that children who were extubated early had a significantly shorter duration of ICU treatment.

Furthermore, children in the OTE group had significantly less volume shifts intraoperatively as well as during the postoperative course. Interestingly, despite the clinically more favorable course in the OTE group, no difference was observed in perioperative markers of inflammation. We can only speculate that the main impact of the surgical procedure was on the expression of systemic inflammatory markers and that the benefit of shorter duration of ventilation and reduced catecholamine therapy is secondary in this regard.

As a surrogate for morbidity, we selected the duration of intensive care therapy and the hospital length of stay. ICU treatment and length of hospital stay were both significantly reduced in OTE patients compared to DET children which is in line with Varghese et al. [26]. For a more detailed consideration of intensive therapy, we focused on the amount and duration of cathecholamine therapy using the VIS score [20]. Patients in the OTE group showed significantly lower VIS scores as well as a shorter duration of catecholamine therapy, indicative of a more stable cardiovascular situation.

Significant demographic and procedural differences between the OTE and DET group need to be acknowledged and raised the question to what extent the favorable postoperative courses were a result of older age, higher somatometric parameters, lower exposition of cardiac surgical stress factors such as surgery and/or CPB time or a result of the early extubation itself. Therefore, we performed propensity score matching for the variables age, weight, complexity of performed procedure using STAT-Sore as well as duration of cardiopulmonary bypass to exclude demographical and procedural differences between groups. After correction for these variables, the outcome parameters duration of intensive care treatment and catecholamine therapy remained statistically significantly reduced in favor of the OTE group, while length of hospital stay did not reach statistical significance.

In one collective of patients, the benefit of early extubation strategy is obvious, based on pathophysiologic considerations. It is well known that positive pressure ventilation has a harmful impact on pulmonary blood flow in patients with Fontan physiology. Therefore, there is strong evidence that in these patients rapid transition to spontaneous breathing improves pulmonary blood flow and thus hemodynamics [15]. It is likely that this might also be the case in patients undergoing Glenn procedure. However, even in these congenital heart defects, where the benefits of early extubation are obvious, the degree to which early extubation is currently performed varies considerably between centers due to preferences or traditions [24]. Our collective included 11 children with Glenn procedure. However, children scheduled for Fontan completion were not included, because this procedure is frequently performed in children over 24 months of age. Of the 12 children who underwent Glenn surgery, 58.33% were extubated in the operating room. Considering the current state of knowledge, it is not clear whether good initial conditions make early extubation possible and subsequently lead to positive outcome, or if early extubation itself has positive impact on clinical course. It could be assumed that especially the infants with low individual and operative risk factors are suitable for early extubation and therefore may have a favorable perioperative course. In our study, we found that OTE was very uncommon in newborns and younger infants with an elevated STATS-score. We speculate that these children in particular represent a special risk collective. However, there were also infants in our study with increased age, increased weight and short bypass times who did not undergo on-table extubation. The question arises why these children were not assigned to an OTE. Another reason for delayed extubation might be a higher STATS-score. Complex cardiac defects or surgical procedures of high complexity were significantly less likely to be extubated on-table. For example, in our study, no infant who underwent a Norwood procedure was extubated on the table. According to a survey of our group in Germany, no on-table extubations are performed after Norwood procedures [17].

Risk classifications such as the STATS-score have been developed to assess this perioperative risk [21]. Early extubations were predominantly performed in STATS-score 1–3, with almost no OTE performed in groups 4 and 5. This reflects the higher level of complexity of the operations in these categories as well as the higher disease burden of the patients. Some physicians do not recommend early extubation in neonates due to the relatively horizontal alignment of the ribs, weak intercostal muscles, narrow subglottic portion, and post-anesthetic apnea [27,28]. In order to consider the influence of possible confounding variables, we performed a propensity score matching, which adjusted for the variables age, weight, complexity of performed procedure using STAT-Sore, as well as duration of cardiopulmonary bypass and considered the nearest neighbor. After matching, both groups were comparable. Outcomes regarding catecholamine therapy, and ICU LOS remained significantly in favor of the OTE group. However, especially in pediatric cardiac surgery, it is difficult to assess the perioperative risk exclusively due to the surgical procedure performed or the underlying heart defect. Within a single diagnosis of a congenital heart defect, such as a ventricular septal defect, there might be a wide range of possible conditions with very different complications and courses. We intended to address this by considering confounding variables, but we are aware that this will remain incomplete in this complex patient population. Nevertheless, our study showed that OTE is associated with a favorable perioperative course. Although in this study, we do not address the adverse effects of prolonged sedation and mechanical ventilation in a particularly vulnerable patient population, the adverse effects of prolonged sedation and prolonged mechanical ventilation in infants are obvious and should therefore be prevented whenever possible.

## 5. Conclusions

Our study indicates that, in pediatric cardiac surgery, OTE is safe and associated with a favorable postoperative course, including fewer catecholamine requirements and shorter duration of ICU therapy.

## Figures and Tables

**Figure 1 jcm-11-05186-f001:**
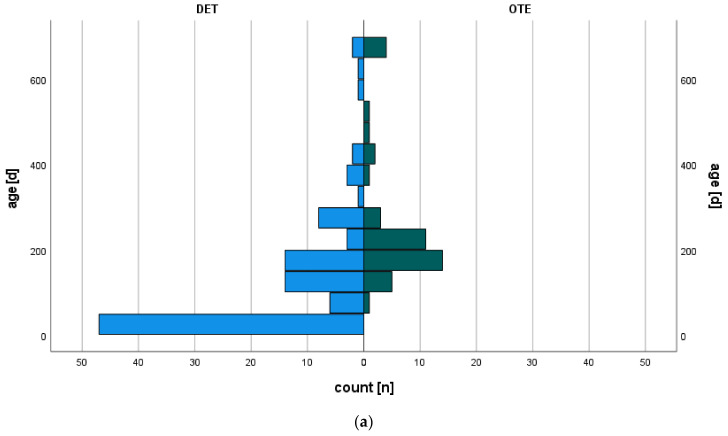
Population pyramid illustrates the distribution of patients in relation to (**a**) age (days of life); (**b**) body weight (kg); (**c**) height (cm). Blue bars represent delayed extubation group (DET) versus on-table extubation (OTE) green bars.

**Figure 2 jcm-11-05186-f002:**
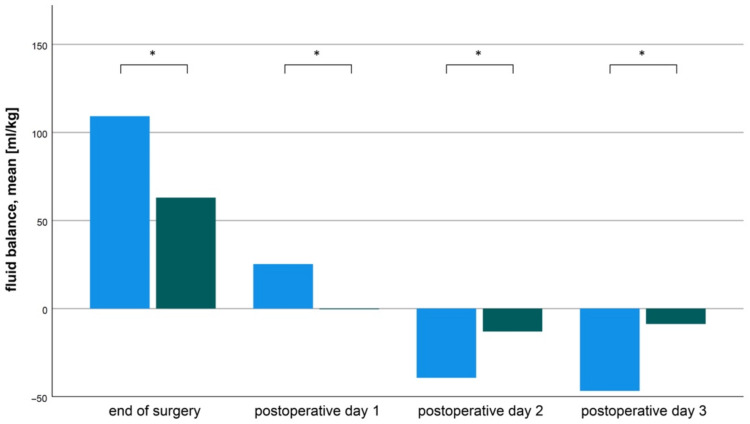
Perioperative fluid balance analysis revealed that the patients in the delayed extubation DET) group had significantly higher fluid shifts than the patients in the on-table extubation (OTE) group. In the diagram the mean fluid balance is shown. The columns represent the mean fluid balance at the time of end of surgery, 1st 2nd and 3rd postoperative day. Blue columns represent the DET group, green columns represent the OTE group. All differ significantly * *p* < 0.0001.

**Figure 3 jcm-11-05186-f003:**
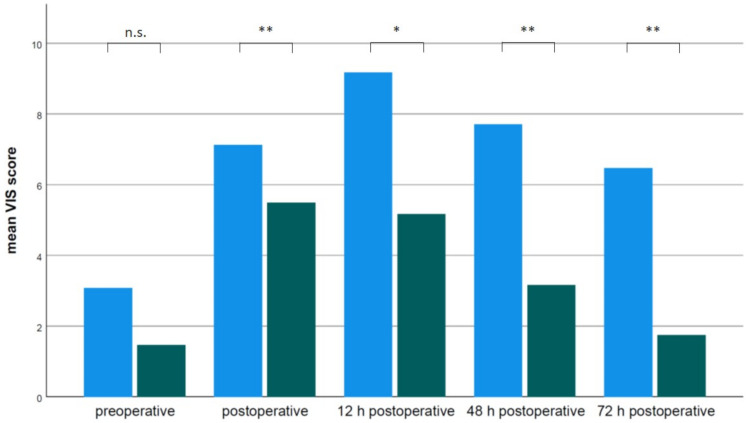
Mean vasoactive-inotropic score (VIS) in delayed extubation (DET) group (blue bars) and on-table extubation (OTE) group (green bars) preoperative, postoperative, 12 h postoperative, 48 h postoperative, 72 h postoperative. n.s. not significant, * *p* < 0.05, ** *p* <0.001.

**Figure 4 jcm-11-05186-f004:**
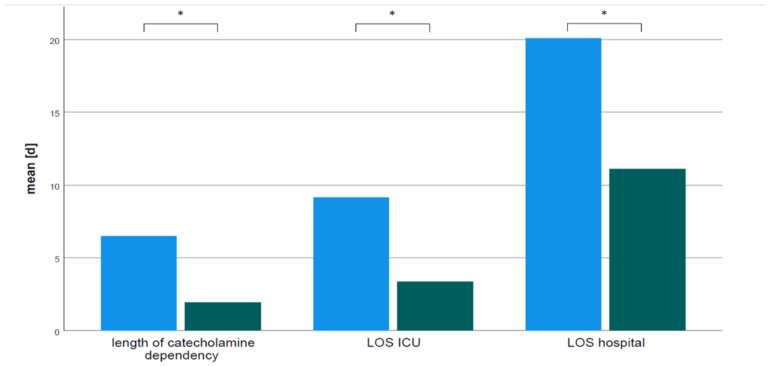
Mean duration of catecholamine dependency, intensive care therapy LOS ICU), and length of hospital stay (LOS hospital) in delayed extubation (DET) group (blue bars) and on-table extubation (OTE) group (green bars). All differ significantly * *p* < 0.0001.

**Figure 5 jcm-11-05186-f005:**
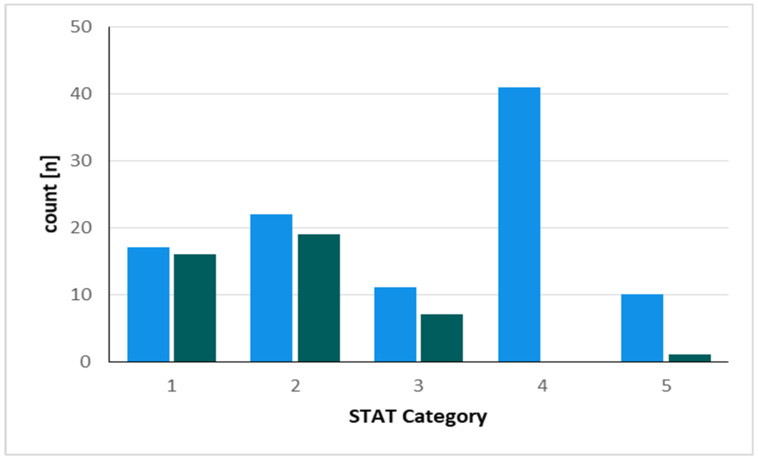
The distribution of cases according to the Society of Thoracic Surgeons-European Association for Cardio-Thoracic Surgery Mortality (STAT) category is shown. Blue columns represent the absolute number of cases in the DET group, the green columns represent the cases in the OTE group.

**Figure 6 jcm-11-05186-f006:**
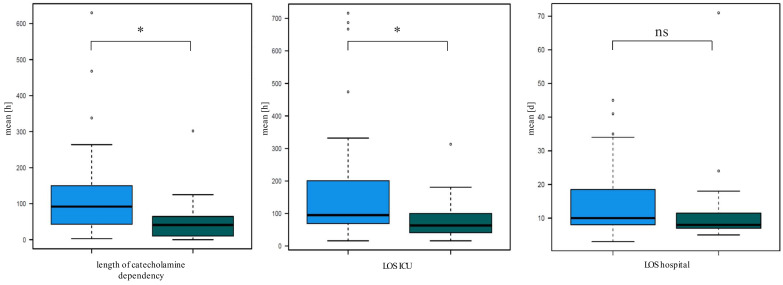
Duration of catecholamine dependency (*p* < 0.001), duration of intensive care therapy (*p* < 0.001), and length of hospital stay (*p* = 0.095) in delayed extubation DET group (blue bars) and on-table extubation OTE group (green bars) after propensity score matching, n.s. not significant, * *p* < 0.05.

**Table 1 jcm-11-05186-t001:** Perioperative markers of inflammation.

Variable	Total		DET		OTE			
	Mean	SD	Mean	SD	Mean	SD	*p*-Value	Missing
Leukocytes preOP [g/L]	10.5	3.5	10.7	3.5	10.1	3.3	0.341	6
Leukocytes postOP [g/L]	12.1	4.4	12.5	4.8	11.4	3.1	0.365	0
Leukocytes postOP day 1 [g/L]	12.1	4.4	12.5	4.8	11.3	3.1	0.261	3
Leukocytes postOP day 2 [g/L]	13.6	9.3	13.0	4.2	14.9	15.8	0.627	2
Leukocytes postOP day 3 [g/L]	10.2	3.7	10.2	3.7	10.3	3.7	0.710	4
CRP preOP [mg/dL]	3.5	11.7	4.3	13.9	1.7	2.7	0.277	8
CRP postOP day 1 [mg/dL]	24.5	25.2	24.4	28.3	24.9	16.4	0.313	14
CRP postOP day 2 [mg/dL]	59.9	48.5	63.4	51.1	52.2	41.6	0.212	8
CRP postOP day 3 [mg/dL]	64.6	60.3	69.6	62.1	53.2	54.8	0.082	13

DET delayed extubation time, OTE on-table extubation, preOP preoperative, postOP postoperative, C-reactive protein (CRP).

## Data Availability

The data presented in this study are available on request from the corresponding author.

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
