# Peer review of "Effects of on-Table Extubation after Pediatric Cardiac Surgery"

_jcm, 2022, doi:10.3390/jcm11175186_

Round 1
Reviewer 1 Report
This is a very interesting paper that highlights the need for additional study of this important topic. It would be helpful to better understand the types of surgeries performed in each of the groups rather than just the STAT category breakdown. For example, in the discussion the authors highlighted how many of the patients were undergoing a Glenn operation, but no other specifics are provided, perhaps a table would be sufficient.
Author Response
This is a very interesting paper that highlights the need for additional study of this important topic. It would be helpful to better understand the types of surgeries performed in each of the groups rather than just the STAT category breakdown. For example, in the discussion the authors highlighted how many of the patients were undergoing a Glenn operation, but no other specifics are provided, perhaps a table would be sufficient.
Thank you very much for pleasant evaluation. We absolutely agree with the reviewer in that additional studies are required to investigate the impact of anesthesia and on-table extubation to evaluate its impact on enhanced recovery after surgery in this special group of patients. As already mentioned in the manuscript the distribution of procedures differs between groups with most procedures been included in both treatments, while some only seen in the delayed extubation group. Propensity score matching has been performed to account for these differences.
According to the Reviewers suggestion we incorporated a table showing specific procedures in each treatment into supplemental data and included the following statement “Pathologies differed significantly between OTE and DET groups with some procedures only extubated after admission to ICU (Supplemental 1).” Into the result section.
Finally, we would like to thank the reviewer for the time and effort invested into our manuscript.

Reviewer 2 Report
Thank you for the opportunity to review your manuscript. This is an interesting patient series but I fear it really does not add any substantial new information to the existing body of knowledge on this topic. The major consideration is the large number of publications surrounding this, I find myself in an unusual position that I cannot decide how to review this article.
The subject is very relevant in the modern era of anesthesia and intensive care. We are practicing OTE at our centre with about 60% of the cohort being extubated on table. We use thoracic epidural analgesia for postop pain control. However, there are different approaches to fast tracking. And yet there are large
academic centers of excellence for congenital surgery who simply do not consider early extubation as a viable option, and they also have excellent outcomes.
There are points to consider:
Ø the most difficulty I have with this paper is the descriptions of methodology.
Ø The anaesthesia and post op ICU protocol is not elaborated, in terms of pain relief and sedation of baby (with lot of lines and drains in situ) and mode of postop oxygenation,
Ø Whats the reintubation rate and ICU complication?
Ø The patients who were extubated on table were definitely stable in your study (also evident from demographic and intraop characteristics) making their outcome variables including duration of ICU treatment, catecholamine dependencies, and hospital LOS, favorable. It is expected, it’s a relation and not effect. Therefore statement “ Our results suggest, that if on-table extubation can 31 be performed, this predicts a favorable postoperative course” does not fit.
Ø Classification based on Diagnosis and STAT score/RACHS score not mentioned.
Author Response
Thank you for the opportunity to review your manuscript. This is an interesting patient series but I fear it really does not add any substantial new information to the existing body of knowledge on this topic. The major consideration is the large number of publications surrounding this, I find myself in an unusual position that I cannot decide how to review this article.
To begin with we would like to thank the Reviewer for ehr time and interest on the topic of the manuscript. Enhanced recovery is extensively investigated in perioperative medicine including pediatric cardiac surgery. We searched the keywords “on table extubation” and “pediatric cardiac surgery” in the medline with no further restrictions and received a total of 19 results. In most publication on table extubation was mentioned beside the investigated topic. A total of four publications investigated on table extubation in pediatric cardiac surgery with a focus on risk assessment, practice survey, pro con, and implementation in the developing world. There is an ongoing discussion with pro and cons. The herein presented manuscript aimed to evaluate the safety and impact on the postoperative course which within a broad spectrum of pediatric cardiac procedures we could not find in the literature.
The subject is very relevant in the modern era of anesthesia and intensive care. We are practicing OTE at our centre with about 60% of the cohort being extubated on table. We use thoracic epidural analgesia for postop pain control. However, there are different approaches to fast tracking. And yet there are large academic centers of excellence for congenital surgery who simply do not consider early extubation as a viable option, and they also have excellent outcomes.
We completely agree with the Reviewers comments. Performing OTE with 60% of the cases is really impressive. At our center we’re not quite there yet. Thoracic epidural anesthesia is currently a hot topic and its usage also differs highly between centers. Unfortunately, due to potential complication it’s not performed at our center for pediatric cardiac surgery. However, in response to a recent report from Schmehil and colleagues about their experience with epidural catheter placement in pediatric cardiac surgery there was an editorial by Kussman concluding that larger prospective studies are required to demonstrate it's safety and effectiveness. Hopefully we can get this implemented when safety concerns are excluded.
Also, we agree with the reviewer that you can receive excellent results at centers that do not perform on table extubation. However, these centers you are referring to most likely have a protocol implemented where patient are extubated early after admission to ICU. In our opinion there are various procedures that can be performed and protocols that can be followed to improve the outcome after pediatric cardiac surgery. The aim of our study was to assess the effects of OTE and to proof its safety.
There are points to consider:
Ø the most difficulty I have with this paper is the descriptions of methodology.
To acknowledge differences between groups propensity score matching including age, weight, complexity of performed procedure using STAT-Sore as well as duration of cardiopulmonary bypass was used to account for potential cofounders between groups. The sample size of the PSM was 86, to each of the 43 cases we matched 43 controls following a nearest neighbor matching algorithm.
To provide complete transparency: This is the output of the R function matchit that was used for this procedure (variable names: Alter = age, Gewicht = weight, STAT.Mortality.Category = STAT score, HLM_mm = duration of cardiopulmonary bypass):
A matchit object
- method: 1:1 nearest neighbor matching without replacement
- distance: Propensity score
- estimated with logistic regression
- number of obs.: 144 (original), 86 (matched)
- target estimand: ATT
- covariates: Alter, Gewicht, STAT.Mortality.Category, HLM_mm
Summary of Balance for All Data:
Means Treated Means Control Std. Mean Diff. Var. Ratio eCDF Mean eCDF Max
distance 0.5353 0.1978 1.5130 1.0322 0.3669 0.6387
Alter 267.2558 127.4752 0.8595 1.1997 0.3069 0.5807
Gewicht 7.0284 4.9484 1.2812 0.7571 0.3219 0.5644
STAT.Mortality.Category 1.8605 3.0495 -1.3804 0.4345 0.2378 0.4817
HLM_mm 118.9070 162.8911 -0.8132 0.7724 0.2369 0.4610
Summary of Balance for Matched Data:
Means Treated Means Control Std. Mean Diff. Var. Ratio eCDF Mean eCDF Max Std. Pair Dist.
distance 0.5353 0.3821 0.6870 0.9741 0.1187 0.4651 0.6937
Alter 267.2558 214.9070 0.3219 0.9188 0.1292 0.3023 0.7718
Gewicht 7.0284 6.1377 0.5486 0.6577 0.1435 0.3953 1.0130
STAT.Mortality.Category 1.8605 1.9302 -0.0810 0.7283 0.0233 0.0698 1.0529
HLM_mm 118.9070 136.0000 -0.3160 0.9404 0.1411 0.3488 1.0479
Percent Balance Improvement:
Std. Mean Diff. Var. Ratio eCDF Mean eCDF Max
distance 54.6 17.2 67.6 27.2
Alter 62.5 53.5 57.9 47.9
Gewicht 57.2 -50.6 55.4 29.9
STAT.Mortality.Category 94.1 62.0 90.2 85.5
HLM_mm 61.1 76.2 40.5 24.3
Sample Sizes:
Control Treated
All 101 43
Matched 43 43
Unmatched 58 0
Discarded 0 0
Here the performed graphical assessment of the performance of the matching:
Ø The anaesthesia and post op ICU protocol is not elaborated, in terms of pain relief and sedation of baby (with lot of lines and drains in situ) and mode of postop oxygenation,
The following paragraph has been included into the method section providing more detailed information regarding to the post surgical analgesia and sedation standard: “In OTE group patients received piritramide (0.2 mg/kg) before propofol and remifentanil infusions were terminated at the end of the surgery. Subsequent analgesia was performed using piritramide (0.05 mg/kg every 4 hours) and paracetamol (10 mg/kg every 6 hours). Medications were given oral beginning on postoperative day 2. For DET group at the end of the surgery patients were sedated during transfer and admission on ICU using rem-ifentanil (10–20 mcg/kg/h) and propofol (5-10 mg/kg/h). Piritramide (0.05 mg/kg every 4 hours) and paracetamol (10 mg/kg every 6 hours) were started before extubation and termination of remifentanil and propofol infusions. Sufentanil (1-3 mcg/kg/h), midazolam 0.1-0.3 mg/kg/h), and ketamin (1 mg/kg) were given patients requiring prolonged sedation. Additional analgesics was given if patients showed signs of pain or stress.”
Ø Whats the reintubation rate and ICU complication?
The following paragraph hat been revised and included into the result section:” reintubation during the course of postoperative intensive care treatment occurred in 1 OTE patient due to re-surgery for bleeding complication and 10 DET patients all due to respiratory complications.”
Ø The patients who were extubated on table were definitely stable in your study (also evident from demographic and intraop characteristics) making their outcome variables including duration of ICU treatment, catecholamine dependencies, and hospital LOS, favorable.
We absolutely agree with the Reviewers comment that patients in OTE group were stable. To compare between OTE and DET group we performed propensity score matching followed by "nearest neighbor" procedures for similar cases as described in the method section. Further details regarding to statistical tests are presented in the response to the Reviser’s first comment.
It is expected, it’s a relation and not effect. Therefore statement “Our results suggest, that if on-table extubation can be performed, this predicts a favorable postoperative course” does not fit.
The Reviewer is addressing the key question if a favorable postoperative course is an effect or related to OTE. We agree that we cannot prove the effect and therefore our statement is an overinterpretation. A randomized trial would be required to prove the relation of OTE on postoperative outcome. Accordingly, we revised the sentence to “Our results suggest, that if on-table extubation after pediatric cardiac surgery is feasible and in our cohort was associated with a favorable postoperative course.”
Ø Classification based on Diagnosis and STAT score/RACHS score not mentioned.
We have included the specific procedures performed in OTE and DET groups into the supplemental data. Based on these information respective STAT-Scores were calculated. We also assessed the RACS score.
Statistical analyses based on this score showed similar results. However, we would preferer not to include both scores into our manuscript because this might potentially be confusing to the reader.
Finally, we would like to thank the Reviewer for his suggestions. We feel that incorporating his valuable comments substantially improved the impact of our manuscript and hope that he is satisfied with our response.

Round 2
Reviewer 2 Report
Article has been well revised.